# Learn to Adapt Parametric Solvers Under Incomplete Physics

**Armand Kassaï Koupaï[1], Yuan Yin[1], Patrick Gallinari[1,2]**
[1]Sorbonne Université, CNRS, ISIR, 75005 Paris, France
[2]Criteo AI Lab, Paris

## Abstract

Modelling physical systems when only partial knowledge of the physics is available is a recurrent problem in science. Within this context, we consider hybrid models that complement PDE solvers, providing incomplete physics information, with NN components for modelling dynamical systems. A critical challenge with this approach lies in generalising to unseen environments that share similar dynamics but have different physical contexts. To tackle this, we use a meta-learning strategy for learning hybrid systems, that captures context-specific variations inherent in each system, enhancing the model's adaptability to generalise to new PDE parameters and initial conditions. We emphasise the advantages of adaptation strategies compared to a pure empirical risk minimisation approach, the superiority of the solver-neural network combination over soft physics constraints, and the enhanced generalisation ability compared to alternative approaches.

## 1 Introduction

Augmenting physical principles with data-driven methodologies has emerged as a promising approach for modelling dynamical systems and solving partial differential equations (PDEs). This can be achieved classically by incorporating physical constraints within a loss function (Raissi et al., 2019; Li et al., 2023). However, this often leads to ill-posed optimisation problems, and incorporating hard constraints is usually more efficient. Hybrid systems that combine PDE solvers with NN components do exactly that. This idea has been explored, for example, for accelerating simulations by enhancing low-resolution solvers with neural networks (NNs) (Kochkov et al., 2021; Um et al., 2021) or for complementing partially known dynamics (Yin et al., 2021; Tathawadekar et al., 2023). We consider here this hybrid setting for solving parametric PDEs, i.e. modelling physical systems subject to varying dynamics due to changes in equation coefficients, initial conditions (IC), boundary conditions (BC), and forcing terms.

A key problem with models leveraging data-driven components is the generalisation issue: these models fail to generalise to unseen contexts or parameters. This is currently addressed through a classical empirical risk minimisation (ERM) practice, by sampling observations or simulations across the parameters distribution (Brandstetter et al., 2023; Takamoto et al., 2023). However, given the complexity and the diversity of observable dynamics in physical systems together with limited data availability, these methods primarily interpolate in a small neighbourhood within the training parameters distribution, lacking generalisation to unseen conditions outside the training ones. Our claim is that empirical risk minimisation is not well-suited for modelling the complexity of physical dynamics. Instead, we advocate for an alternative approach that emphasises systems capable of rapidly adapting to novel environments. Our framework can be viewed as a meta-learning approach for rapidly adapting the data-driven components of a hybrid model. Our contributions are as follows: • We introduce an **adaptation strategy for hybrid Physics-aware neural parametric PDE solvers** providing enhanced extrapolation abilities compared to alternative approaches. • This allows **low-cost adaptation** to new situations with only a small number of observed data. • We evaluate the benefits of this approach across various datasets, **encompassing both-in-distribution and out-of-distribution scenarios**. • We compare this hard physical constrained approach to soft constraint incorporation.

## 2 METHODOLOGY

### 2.1 PROBLEM DESCRIPTION AND NOTATIONS

We consider general dynamics driven by equations of the form:

$$\frac{\mathrm{d}u(x,t)}{\mathrm{d}t} = f(u(x,t)), \tag{1}$$

with $u(x,t)$ the PDE solution at time $t \in I \subset [0,\infty)$ at position $x$ in space $\Omega$, $f \in \mathcal{F}$ maps the PDE solution $u$ to its temporal derivatives. Our objective is to model the temporal evolution of a deterministic spatio-temporal phenomenon. We posit the observation of similar phenomena occurring in diverse environments $e$, each leading to a distinct dynamics denoted $f^e$. Specifically, we assume a dynamical system modelled by a parametric PDE, where the form of the PDE is shared across environments, but the parameter values vary among them. We also assume that the physics is partially known and provided as an explicit PDE. The dynamics equation for a given environment can thus be rewritten as:

$$\frac{\mathrm{d}u^e(x,t)}{\mathrm{d}t} = G(f^e(u(x,t)), r^e(u(x,t))), \tag{2}$$

$f^e$ and $r^e$ respectively represents the known and unknown physics of environment $e$ and $G$ is a simple function combining the known and unknown components of the PDE.

For training, we assume the observation of a set of trajectories for each environment $e$, each defined by an IC $u_0$. We define $E_{tr}$ and $E_{ad}$, respectively the set of environments used to train the model and to adapt to during inference. In each training environment, a small amount of $N$ trajectories is available to train the model and composes a training set $\mathcal{D}_{tr}^e = \{u_i(x,t)\}_{i=1}^N$. During inference, we adapt our model using only a single trajectory per environment denoted as $\mathcal{D}_{ad}^e$. We define the MSE loss over training set $\mathcal{D}_{tr}^e$ (more details in Appendix C):

$$\mathcal{L}(\theta, \mathcal{D}_{tr}^e) = \sum_{j=1}^N \int_{t \in I, x \in \Omega} \|(g_\theta(u_j(x,t)) - G(f^e(u_j(x,t)), r^e(u_j(x,t))\|_2^2 \mathrm{d}x\mathrm{d}t \tag{3}$$

with $g_\theta$ an approximation function for the true dynamics. Our goal is twofold: first, to learn a function $g_\theta$ capable of generalising to new trajectories from a set of known functions $f^e \in \mathcal{F}_{tr}$, where $e \in E_{tr}$; and second, to efficiently adapt to new environments $e \in E_{ad}$ during which $E_{tr} \cap E_{ad} = \emptyset$. The dynamics in the unseen environments are defined by functions $f^e \in \mathcal{F}_{ad}$ not encountered during training. From a meta-learning view, environments can be viewed as tasks we want to generalise to.

### 2.2 HYBRID FORMULATION FOR LEARNING DYNAMICS

In equation 2, the function $G$ is typically unknown. We assume a simple form and consider two instances for decomposing the known and unknown parts of the PDE in our approximation model $g_\theta$. The first one is a simple additive decomposition:

$$g_\theta = g_p + g_c(\theta) \tag{4}$$

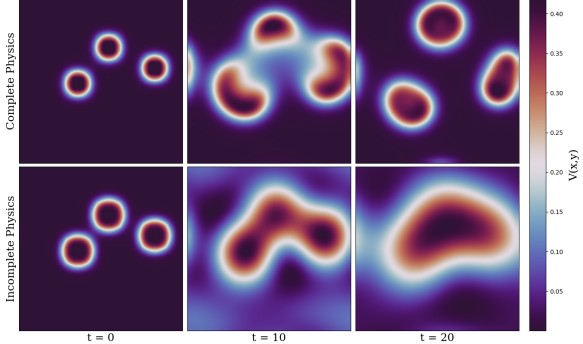

Figure 1: Complete (up) and Incomplete (down) Physics for Gray-Scott PDE

where $g_p \in \mathcal{G}_p$ encodes the incomplete physical knowledge and $g_c \in \mathcal{G}_c$ is the data-driven term complementing $g_p$. It is assumed here that $g_p$ is incomplete but is an exact component of the complete PDE and $g_c(\theta)$ is then modelling a residual term. Although simple, this decomposition can be shown to cover a large variety of situations (Yin et al., 2021). We illustrate in figure 1 the difference between a complete and incomplete trajectory for Gray-Scott PDE a reaction-diffusion system where the missing part corresponds to the reaction terms.

As a second instance, we considered the frequent case where a solver operating at a low resolution solves a simplified dynamics and has to be complemented to approximate the effects of unresolved

small scales onto larger ones (Belbute-Peres et al., 2020; Kochkov et al., 2021; Um et al., 2021). This framework is known as a closure model and is used for example in Large eddy simulation (LES), illustrated in figure 2. By operating on larger scale dynamics than direct numerical simulation (DNS), LES offer a balanced compromise between computational cost and accuracy (Kochkov et al., 2021). Since the closure model operates on the low resolution dynamics model, a better decomposition is:

$$g_\theta = (g_c \circ g_p)(\theta) + g_p \tag{5}$$

Here the data-driven part $g_c$ will model the closure for the incomplete physical component $g_p$. Practically $g_c$ will be trained to complement $g_p$ in order to approximate LES with closed terms trajectories as ground-truth. Starting from time $t_0$, we use an auto-regressive formulation to compute the full trajectory, implemented by a Neural ODE (Chen et al., 2018) which predicts the state $u_{t+\tau}$ as $u_{t+\tau} = u_0 + \int_{t_0}^{\tau} g_\theta(u(\tau)) \mathrm{d}\tau$.

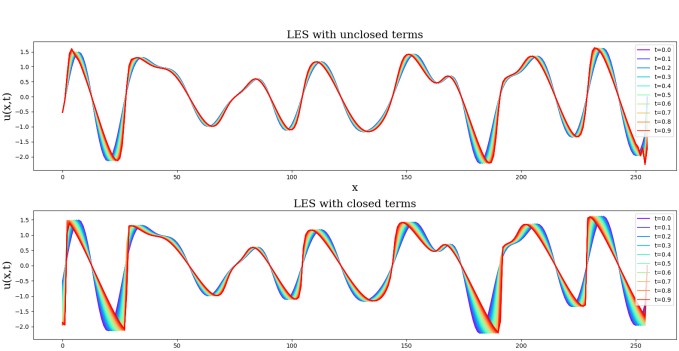

Figure 2: Unclosed and closed LES for Burgers equation

## 2.3 A TWO STAGE FRAMEWORK

Our objective is to train a dynamical model $g_\theta$ on a set of environments and subsequently adapt swiftly and effectively to new environments, as in Kirchmeyer et al. (2022). To achieve this, we introduce two types of parameters for $g_\theta$: $\theta \triangleq \{\theta^e, \theta^a\}$ with $\theta^e$ the environment-specific and $\theta^a$ the environment-agnostic parameters. The proposed optimisation problem is formulated as follows:

**Algorithm 1:** Training Pseudo-code
**Input:** $\{\mathcal{D}_{tr}^e\}_{e \in E_{tr}}$ with $\#\mathcal{D}_{tr}^e = N$;
$\quad\quad$ $\theta = \{\theta^a, \theta^e\}$, random init for $\theta^a$ and $\theta^e$
**while** *no convergence* **do**
$\quad$ $\theta \leftarrow \theta - \eta \nabla_\theta (\sum_{e \in E_{tr}} \mathcal{L}(\theta, \mathcal{D}_{tr}^e) + \lambda_{\theta^e} \|\theta^e\|)$
**end**

$$\min_{\theta^e, \theta^a} \sum_{\mathcal{D}_{tr}^e \in E_{tr}} \mathcal{L}(\{\theta^e, \theta^a\}, \mathcal{D}_{tr}^e)$$

$$\text{subject to } \theta^e = \arg\min_{\theta^e} \sum_{\mathcal{D}_{tr}^e \in E_{ad}} \mathcal{L}(\theta^e, \mathcal{D}_{tr}^e) \tag{6}$$

Our meta-learning strategy operates in two steps: Training and Adaptation. Then, it's applied to a new environment, inferring and adapting based on a single trajectory from that environment.

**Training stage** The objective is to learn the best environment-agnostic and environment-specific parameters. Ideally, we want the environment-agnostic parameters to capture most of the dynamics and only use the environment-specific parameters to capture information not shared across environments. We impose a locality constraint such that $\theta^e$ remains small:

$$\min_{\theta = \theta^e + \theta^a} \|\theta^e\| \text{ s.t. } \forall u^e(t) \in \mathcal{D}_{tr}^e, \frac{\mathrm{d}u^e(t)}{\mathrm{d}t} = G(g_p, g_c(\theta^e + \theta^a))(u^e(t)) \tag{7}$$

**Adaptation stage** All parameters except environment-specific parameters are fixed. We want fast and accurate adaptation to new environments: we employ a linear hyper-network during training and adaptation stage that operates in a low-dimensional space, facilitating low-cost adaptation:

$$g_\theta = G(g_p, g_c(\theta^a + W^a c^e)) \tag{8}$$

here $\theta^e = W^a c^e$, with $W^a = (W_1, ..., W_{d_c}) \in \mathbb{R}^{d_\theta \times d_c}$, a shared weight matrix learnt during training across environments and $c^e$, a context vector learnt during adaptation. Adaptation to new environments thus only requires learning a small dimensional code $c^e$.

## 3 EXPERIMENTS

Table 1: **Generalisation results for out-domain environments for in and out range time horizon** - Test results. Metrics in MSE.

| Type ↓ | Dataset → Model ↓ | Gray-Scott | | Pendulum | | Lotka-Volterra | | Burgers | |
|---|---|---|---|---|---|---|---|---|---|
| | | In-t | Out-t | In-t | Out-t | In-t | Out-t | In-t | Out-t |
| Data-driven | CoDA | 1.08e-3 | 9.07e-3 | 3.37e-3 | 1.23e-3 | 1.54e-3 | 5.14e-1 | 3.92e-3 | 2.57e-2 |
| | CAVIA | 1.21e-2 | 1.24e-2 | 6.36e-3 | 4.06e-3 | 1.07e-2 | 4.29e-1 | 6.49e-3 | 4.81e-2 |
| Hybrid | CoDA + Phys loss | 3.36e-3 | 8.62e-3 | 1.99e-1 | 2.03 e-1 | 1.42 | 1.7 | 3.67e-2 | 2.28e-1 |
| | Ours | **3.07e-5** | **7.17e-4** | **7.95e-6** | **5.23e-6** | **1.63e-4** | **9.46e-3** | **1.96e-4** | **5.86e-3** |
| Model-driven | Incomplete PDE Solver | 5.55e-1 | 6.10e-1 | 8.51 | 6.23e1 | 9.34 | 1.35e1 | 2.70e-2 | 1.50e-1 |

---

**Algorithm 2:** Adaptation Pseudo-code

**Input:** $\{\mathcal{D}^e_{ad}\}_{e \in E_{ad}}$ with $\#\mathcal{D}^e_{ad} = 1$; fixed $W^a \in \mathbb{R}^{d_\theta \times d_c}$ and $\theta^a \in \mathbb{R}^{d_\theta}$, $c^e = \mathbf{0} \in \mathbb{R}^{d_c}$

**while** *no convergence* **do**

$\quad | \quad c^e \leftarrow c^e - \eta \nabla_{c^e} (\sum_{e \in E_{ad}} \mathcal{L}(c^e, \mathcal{D}^e_{ad}))$

**end**

---

Our approach is validated on representative ODEs/PDEs for the forecasting task. We assess the model performance on two key aspects. • **In-domain generalisation**: the model capability to predict trajectories defined by unseen IC on all training environments $e \in E_{tr}$. • **Out-of-domain generalisation**: the model ability to adapt to a new environment $e \in E_{ad}$ by predicting trajectories defined by unseen IC. We also report *in-range time error (In-t)* and *out-of-range time error (Out-t)* to highlight the model's ability to predict dynamics within and outside the time horizon it has been trained on.

**Setup** • **Datasets** We used two ODEs and two PDEs to assess the performance of our method: Lotka-Volterra (LV, Lotka (1925)) and damped pendulum for the ODEs, 1D Burgers (Basdevant et al., 1986) and 2D Gray-Scott (GS, Pearson (1993)) equations for the PDEs. For all the datasets, we assume that the system is only partially known (details on the known/ unknown terms are provided in Appendix D for lack of space). • **Baselines** We compare our approach with data-driven, hybrid and model-driven baselines. CoDA Yin et al. (2022) and CAVIA Zintgraf et al. (2019) are data-driven meta-learning frameworks for fast adaptation to new environments. We also extend CoDA into a hybrid version by adding to the original data-driven loss a physics-informed loss representing the known physics. This represents a soft constraint approach for integrating physics into a data-driven model, whereas our hybrid approach, which directly incorporates a PDE into the model, corresponds to a hard constraint approach. We also compare to the incomplete physical solver's performance when used alone. • **Implementation** We used MLPs for Pendulum and LV datasets and resolution-dependent ConvNets for Burgers and GS. We use a RK4 solver for the neural ODE and context vectors of dimension 2 for all datasets (more details are available in appendix E).

**Main results** • **In-Domain and Out-Domain generalisation** We report out-domain and in-domain generalisation results in table 1 and 2 (in appendix A), respectively. For both in-domain and out-domain, we outperform the baselines across all datasets sometimes with 1 or 2 orders of magnitude, showcasing the effectiveness of our adaptation approach. CoDA enhanced with physical loss performs worse than the original CoDA both for in and out-domain. Using an incomplete physical loss is not a viable alternative for building hybrid systems, while directly incorporating PDE solvers as a hard constraint proves effective. Besides, as noted by several authors, optimising physics loss from collocation points is an ill-posed problem. The incomplete PDE solver low performance on all the datasets highlights the benefits of the hybrid + adaptation approach. • **In-range and out-range temporal generalisation** Results for both in-range and out-range time horizons are reported in tables 1 and 2 (in appendix A). As all models have been trained only on *In-t* horizon, error accumulates over time and leads to lower performance outside the training horizon. However, our framework consistently exhibits the best score across all datasets when extrapolating outside the training horizon. • Ablation in Table 3 (Appendix A) presents the model performance without the adaptation mechanism. Here, the model learns a single corrective term for all training environments and directly infers out-domain environments without adaptation. The performance is significantly lower than with adaptation, affirming the essential role of adaptation in modelling complex physics and underscoring the inadequacy of the ERM approach in this context.

## 4  CONCLUSION

Our framework demonstrates the importance of adaptation for generalising to new environments and its superiority to the classical ERM setting. It also highlights the benefits of directly embedding PDE solvers as hard constraints in data-driven models compared to soft loss constraints.

## 5  ACKNOWLEDGEMENTS

We acknowledge the financial support provided by DL4CLIM (ANR-19-CHIA-0018-01), DEEPNUM (ANR-21-CE23-0017-02), PHLUSIM (ANR-23-CE23-0025-02), and PEPR Sharp (ANR-23-PEIA-0008", "ANR", "FRANCE 2030").

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

# A   ADDITIONAL RESULTS

## A.1   IN-DOMAIN RESULTS

We report the results for in-domain generalisation in table 2. Across all datasets, we perform better than all baselines, often by 1 or 2 order of magnitude. Our approach achieves also better performance outside of the temporal horizon it has been trained on, despite a decrease in performance compared to the performance inside of training-horizon, due to the accumulation error of the auto-regressive method.

## A.2   MODEL'S PERFORMANCE WITHOUT ADAPTATION MECHANISM

In order to justify the importance of adaptation in our approach, we also evaluate the performance of our hybrid model but without any adaptation mechanism. This comes down to an ERM approach, where we evaluate our model on data coming different environments. We report the results of this approach in table 3 for both in-domain and out-domain trajectories. While an ERM approach can be used to generalise to the same dynamics it has been trained on, i.e in-domain results, it fails to predict trajectories coming from unseen environments (i.e. out-domain results) compared to adaptation methods. In some cases, an ERM approach achieves worse than an incomplete PDE solver alone, highlighting the importance of adaptation.

Table 2: **Generalisation results for in-domain environments for in and out range time horizon** - Test results. Metrics in MSE.

| Type ↓ | Dataset → | GS | | Pendulum | | LV | | Burgers | |
|---|---|---|---|---|---|---|---|---|---|
| | Model ↓ | In-t | Out-t | In-t | Out-t | In-t | Out-t | In-t | Out-t |
| Data-driven | CoDA | 3.39e-4 | 5.58e-3 | 7.91e-5 | 1.80e-5 | 1.61e-5 | 5.52e-2 | 3.81e-4 | 6.93e-3 |
| | CAVIA | 4.86e-4 | 6.45e-3 | 5.10e-5 | 1.11e-5 | 2.54e-5 | 3.34e-2 | 6.50e-3 | 4.81e-2 |
| Hybrid | CoDA + Phys loss | 3.98e-4 | 6.24e-3 | 7.08e-3 | 1.10e-3 | 3.84e-2 | 4.30e-1 | 4.48e-2 | 2.27e-1 |
| | Ours | **4.39e-5** | **3.39e-4** | **6.90e-8** | **2.17e-8** | **5.07e-6** | **6.09e-4** | **1.77e-4** | **5.82e-3** |
| Model-driven | Incomplete PDE Solver | 5.35e-1 | 5.77e-1 | 4.85 | 4.36 | 4.62 | 4.37 | 1.35e-2 | 8.21e-2 |

Table 3: **Generalisation results of a hybrid model without adaptation** - Test results. Metrics in MSE.

| Model ↓ | Dataset → | GS | | Pendulum | | LV | | Burgers | |
|---|---|---|---|---|---|---|---|---|---|
| | Environment ↓ | In-t | Out-t | In-t | Out-t | In-t | Out-t | In-t | Out-t |
| ERM | In-domain | 4.91e-3 | 2.11e-2 | 9.32e-3 | 3.10e-3 | 9.32e-3 | 6.88e-6 | 4.07e-3 | 1.20e-2 |
| | Out-domain | 3.37e-2 | 7.72e-2 | 1.40e1 | 1.44e2 | 3.79 | 6.48 | 2.27e-2 | 1.49e-1 |
| Incomplete PDE Solver | In-domain | 5.35e-1 | 5.77e-1 | 4.85 | 4.36 | 4.62 | 4.37 | 1.35e-2 | 8.21e-2 |
| | Out-domain | 5.55e-1 | 6.10e-1 | 8.51 | 6.23e1 | 9.34 | 1.35e1 | 2.70e-2 | 1.50e-1 |

# B   RELATED WORK

We review hybrid and generalisation methods for learning physical processes, notably in the context of dynamical systems.

## B.1   HYBRID LEARNING

Purely data-driven approaches have emerged as a new tool for PDE solving (Raissi et al., 2019) and dynamics forecasting (de Bézenac et al., 2019). While flexible and proficient in capturing intricate patterns (Li et al., 2021; Brandstetter et al., 2023; Serrano et al., 2023), they tend to fall short when it comes to generalising to different operational conditions. This limitation arises from a lack of fundamental understanding of the systems they model. To unlock the full potential of deep learning in simulating physical processes, there is a growing need to incorporate physical insights into the deep learning framework.

**Data-driven with soft constraints**   Hybrid methodologies have emerged as a solution, combining machine learning techniques with physics losses (Wang et al., 2021; Li et al., 2023). These hybrid approaches have gained traction for their ability to achieve robust generalisation. Within this framework, neural networks often play a crucial role, modelling specific aspects of conventional PDE solvers. However, when only incomplete knowledge is assumed, minimising an incomplete physic loss is not desirable.

**Correcting PDE discretization error**   Recent years have witnessed the development of deep learning models tailored for accurate simulation of turbulent flows (Kochkov et al., 2021). These models leverage the training of neural networks with differentiable physics, addressing numerical errors inherent in PDE discretization. This highlights the capacity of neural networks to swiftly and effectively correct errors in under-resolved simulations. In our approach, we propose a more general formulation for learning incomplete physics or augmenting simplified PDEs.

**Incomplete physical models**   In alignment with the objectives of our work, some researchers have put forth frameworks aimed at augmenting incomplete physical dynamics with neural network models (Yin et al., 2021; Takeishi & Kalousis, 2021; Tathawadekar et al., 2023). This trend reflects a broader effort within the scientific community to synergy the strengths of physical principles and deep learning for improved modelling and simulation of complex systems. However, none of those works tackle the problem of generalisation to new PDE parameters.

## B.2   GENERALISATION FOR DYNAMICAL SYSTEMS

In the context of dynamical systems, various methods have been explored to address the challenge of out-of-distribution generalisation (Yin et al., 2022). Standard deep learning models excel at learning features aligned with the training data distribution but often struggle when faced with test data distributions significantly divergent from the training data.

**Meta-Learning**   One strategy to address this challenge involves the application of meta-learning approaches (Zintgraf et al., 2019; Wang et al., 2022). These approaches seek to imbue models with general knowledge about the system by exposing them to a diverse set of M tasks, thereby enhancing adaptability to varying input distributions. Recent works have proposed a meta-learning formulation specifically for learning models capable of rapid adaptation to unseen tasks within the context of dynamical systems. We propose a similar approach, but extends to more complex dynamical systems where partial knowledge is assumed (Kirchmeyer et al., 2022; Wang et al., 2022; Park et al., 2023).

**Incorporating PDE parameters**   Recent advancements in the pursuit of robust generalisation have introduced innovative techniques designed to bolster model performance in both long-term (Lippe et al., 2023) and out-of-distribution scenarios. Some of these techniques focus on the parameters of partial differential equations (PDEs) (Takamoto et al., 2023), recognising them as pivotal factors influencing data distribution. Consequently, certain methods endeavour to explicitly encode the true parameter values into the latent space (Fotiadis et al., 2022). However, most of these work only operate on ODEs or assume availability of larger amount of data.

**Generalisation with PINNs**   An alternative approach involves physics-informed neural networks (PINN) (Raissi et al., 2019), which serve as PDE solvers. However, a limitation of PINNs is their inability to generalise to diverse scenarios. Therefore, ongoing research efforts have extended the capabilities of PINNs to facilitate generalisation across various system parameters, ICs and boundary conditions (Huang et al., 2022; Cho et al., 2023). Approaches based on PINNs are data-free and thus assume complete knowledge of the PDE. In scenarios where only partial knowledge is assumed, soft constraints are ineffective compared to our proposed approach.

## C   TRAJECTORY BASED FORMULATION

In practice, $f^e$ is unavailable and we can only approximate it from discretised trajectories. As done in (Kirchmeyer et al., 2022), we use a trajectory-based formulation of Eq. (3). We consider a set of trajectories discretised over a uniform temporal and spatial grid includes $\frac{T}{\Delta t}(\frac{s}{\Delta s})^{d_s}$ states, where $d_s$

Table 4: PDE parameters used in training and adaptation experiments

| PDE | training ODE/PDE parameters | adaptation ODE/PDE parameters |
|---|---|---|
| Damped pendulum | $T_0 \in \{5, 6, 7\}$ $\alpha \in \{0.3, 0.4, 0.5\}$ | $T_0 \in \{4, 9\}$ $\alpha \in \{0.1, 0.6\}$ |
| Lotka-Volterra | $\beta \in \{0.5, 0.75, 1.0\}$ $\delta \in \{0.5, 0.75, 1.0\}$ | $\beta \in \{0.3, 1.125\}$ $\delta \in \{0.3, 1.125\}$ |
| 1D Burgers | $\nu \in \{1e{-}4, 3e{-}4, 5e{-}4, 7e{-}4\}$ | $\nu \in \{1e{-}5, 5e{-}5, 5e{-}3, 7e{-}3\}$ |
| 2D Gray-Scott | $F \in \{0.03, 0.039\}$ $k \in \{0.058, 0.062\}$ | $F \in \{0.025, 0.042\}$ $k \in \{0.050, 0.065\}$ |

is the spatial dimension. $\Delta t$ and $\Delta s$ represent respectively the temporal and spatial resolution. $T$ and $S$ are the temporal horizon and spatial grid size. Our loss writes as:

$$\mathcal{L}(\theta, D_{tr}^e) = \sum_{j=1}^{N} \sum_{k=1}^{(s/\Delta s)} \sum_{l=1}^{T/\Delta T} \|u_j^e(s_k, t_l) - \tilde{u}_j^e(s_k, t_l)\|_2^2$$

$$\text{where } \tilde{u}^e(t_l) = u_0^e + \int_{t_0}^{t_k} g_\theta(\tilde{u}^e(\tau)) \mathrm{d}\tau$$

(9)

$u_j^e(s_k, t_l)$ is the state value in the $j^{th}$ trajectory from environment $e$ at the spatial coordinate $s_k$ and time $t_l \triangleq l\Delta t$. $u^e(t) \triangleq [u(s_1, t), \ldots, u(s_{(S/\Delta s)^{d_s}}, t)]^T$ is the state vector in the $j^{th}$ trajectory from environment $e$ over the spatial domain at time t and $u_0^e$ is the corresponding IC.

## D    DATASET DETAILS

We present the equations and the data generation settings used for all dynamical systems considered in this work. In table 4, we report all ODE and PDE parameters used to generate training and adaptation environments.

### D.1    DAMPED PENDULUM

The ODE represents the motion of a simple pendulum:

$$\frac{d^2\theta}{dt^2} + \omega_0^2 \sin\theta + \alpha\frac{d\theta}{dt} = 0$$

(10)

where $\theta(t)$ is the angle, $\omega_0$ the proper pulsation and $\alpha$ the damping coefficient. In our case where we suppose we only have incomplete knowledge of the phenomenon, we consider the term $\alpha\frac{d\theta}{dt}$ unknown and want to learn it using the data-driven component.

**Data generation**    We generate trajectories coming from different environments represented by specific parameter values for both $\alpha$ and $\omega_0 = (\frac{2\pi}{T_0})^2$ using Runge-Kutta 8 solver. For training, we generated 9 distinct environments, each composed of 16 trajectories on the time horizon $[0, 10]$ with a time step $\Delta t = 0.5$. For adaptation, we evaluate our method on 4 distinct environments defined by parameters unseen during training and trajectories time horizon is $[0, 20]$. Only one trajectory per environment is used to adapt the model to those new dynamics and we evaluate the model's performance on 32 trajectories per environments.

### D.2 LOTKA-VOLTERRA

The system describes the interaction between a prey-predator pair in an ecosystem, formalised into the following ODE:

$$\frac{dx}{dt} = \alpha x - \beta xy \tag{11}$$

$$\frac{dy}{dt} = \delta xy - \gamma y \tag{12}$$

where $x, y$ are respectively the quantity of the prey and the predator, $\alpha, \beta, \delta, \gamma$ define how two species interact. Across all environments, we suppose $\alpha = \gamma = 0.5$, with only $\beta, \delta$ varying across environments. For the physical model, the term $\gamma y$ is supposed unknown.

**Data generation**  We generate trajectories on a temporal grid $\Delta t = 0.5$ using Runge-Kutta 45 solver. For training, we consider 9 different environments, each composed of 4 trajectories, each defined by specific ICs. For adaptation, we adapt our network on 4 new environments using only 1 trajectory per environment. We evaluate the adaptation's performance on 32 trajectories per environment. Trajectories for training have been generated on a time horizon $[0, 10]$ and on $[0, 20]$ for evaluation.

### D.3 BURGERS

Burgers' equation is a nonlinear equation which models fluid dynamics in 1D and features shock formation. We consider the following form of the Burgers equation:

$$\frac{du}{dt} + \frac{du}{dx} = \nu \frac{d^2 u}{dx^2} \tag{13}$$

where $u$ is the velocity field and $\nu$ is the diffusion coefficient.

**Data generation**  For the DNS, we generate complex trajectories using a $5^{th}$ order central difference scheme using Runge-Kutta 45 solver with a time-step $\Delta t = 1\mathrm{e}{-5}$ and $\Delta x = \frac{2\pi}{16384}$. Such trajectories are particularly costly to generate, therefore, we rather use LES. To obtain the ground truth LES trajectories, we apply a spatial filtering operator on the DNS trajectories. We also down-sample the temporal and spatial grid. Therefore, we obtain LES trajectories with a timestep $\Delta t = 1\mathrm{e}{-3}$ and $\Delta x = \frac{1}{256}$. Applying such filtering operator to the Burgers equation leads to the following formulation:

$$\frac{d\bar{u}}{dt} + u\frac{d\bar{u}}{dx} = \nu \frac{d^2 \bar{u}}{dx^2} + \mathcal{R}^{\text{closure}}(\bar{u}, u) \tag{14}$$

where $\mathcal{R}$ represents the unresolved subgrid scales, which are smaller than the resolved scales in the coarse grid. We try to learn these unresolved scales with the data-driven component. For training, we generate 4 environments with 4 trajectories per environment, each corresponding to a different viscosity coefficient. Trajectories have been generated on a temporal horizon $[0, 0.05]$ for training and on $[0, 0.1]$ for evaluation.

### D.4 GRAY-SCOTT

The PDE describes reaction-diffusion system with complex spatiotemporal pattern through the following 2D PDE:

$$\frac{du}{dt} = D_u \Delta u - uv^2 + F(1 - u) \tag{15}$$

$$\frac{dv}{dt} = D_v \Delta v - uv^2 - (F + k)v \tag{16}$$

where $u, v$ represent the concentrations of two chemical components in the spatial domain $S$ with periodic boundary conditions. $D_u, D_v$ denote the diffusion coefficients respectively for $u, v$ and $F, k$ are the reactions parameters. Across all environments, $D_u = 0.2097$, $D_v = 0.105$ are constant. We consider the terms $F(1 - u)$ and $(F + k)v$ unknown and use our framework to learn the missing terms.

**Data generation**  We generate trajectories on a temporal grid with $\Delta t = 10$. $S$ is a 2D space of dimension $32 \times 32$ with a spatial resolution of $\Delta s = 2$. For training, we generate 4 environments with one trajectory per environment defined on a temporal horizon $[0, 200]$. For adaptation, we generated also 4 environments and only use 1 trajectory per environment to adapt to the new dynamics. We then evaluate the framework's performance on 32 new trajectories per environments, defined on a temporal horizon $[0, 400]$.

## E  IMPLEMENTATION DETAILS

The code has been written in Pytorch (Paszke et al., 2019). All experiments were conducted on a single GPU: NVIDIA TITAN Xp with 12 Go.

### E.1  ARCHITECTURE

We implement the dynamical model $g_\theta$ with the following architectures:

- LV and Pendulum: we used MLPs composed of 4 hidden layers of width 64.
- GS: 4-layer 2D ConvNet with 64-channel hidden layers, and $3 \times 3$ convolution kernels.
- BG: 4-layer 1D ConvNet with 64-channel hidden layers, and $7 \times 7$ convolution kernels.

For all networks, we use Swish activation layers. The hyper-network used is a single affine layer NN. We use an RK4 solver for the Neural ODE. For the Gray-Scott equation, we use a temporal step size $\Delta t = 1$, as missing terms are present, using larger steps diverge.

For all datasets, we used context vectors of dimension $d_c = 2$

### E.2  OPTIMISER

For all datasets, we use the Adam optimiser with a learning rate of $10^{-3}$ and $(\beta_1, \beta_2) = (0.9, 0.999)$ for both training and adaptation. For training, we used a learning rate scheduler which reduces the learning rate when the loss has stopped improving. We set the threshold to $0.01$ with a patience of 250 epochs with respect to the training loss. The minimum learning rate is $1e - 5$. During training, we also regularise context parameters and uses a weight coefficient $\lambda_{\theta^e} = 1e - 6$.

### E.3  BASELINES IMPLEMENTATION

For all baselines, we followed the recommendations given by the authors. For the CoDA with a soft constraint baseline, we implemented the physics losses using finite difference method.

## F  QUALITATIVE RESULTS

In this section, we show different visualisation of the predictions made by our framework and we compare them with the trajectories generated by baselines. We report in the figure 3 the predictions made by baselines and our approach on a trajectory from an adaptation environment. In figure 4, we show the prediction made by our approach on a trajectory coming from an enviroment used for adaptation and compare it to baselines.

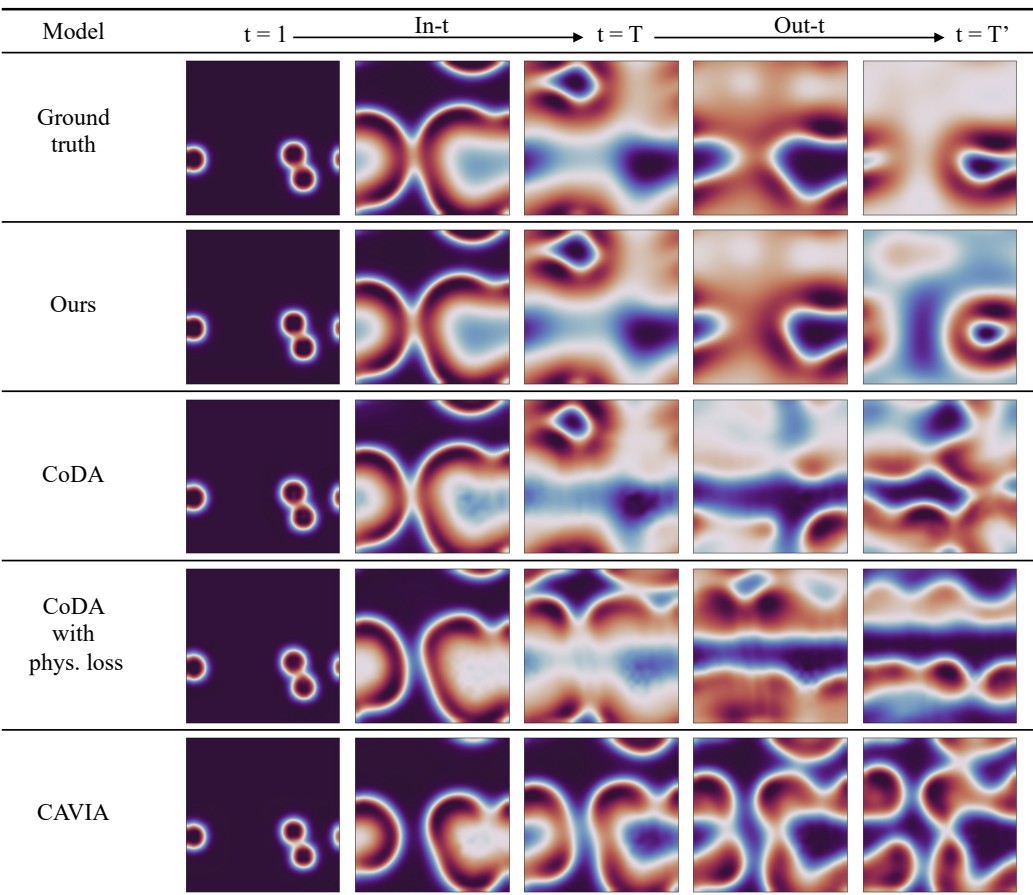

Figure 3: Prediction per frame for our approach on 2D Gray-Scott on the adaptation environment $F, k, D_u, D_v = (0.025, 0, 050, 0.2097, 0.105)$. The trajectory is predicted from t = 0 to t = T'. In our setting, T = 19 and T' = 39.

Figure 4: Prediction of a trajectory on 1D Burgers on the adaption environment $\nu = 5e - 5$

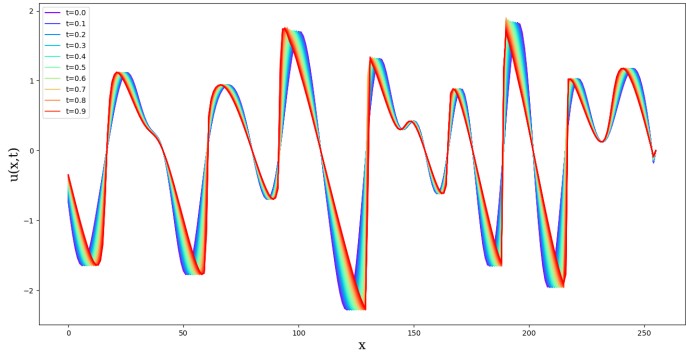

Figure 5: Ground Truth

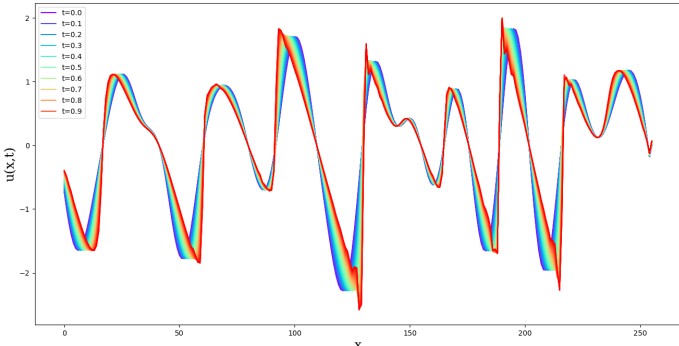

Figure 6: Ours

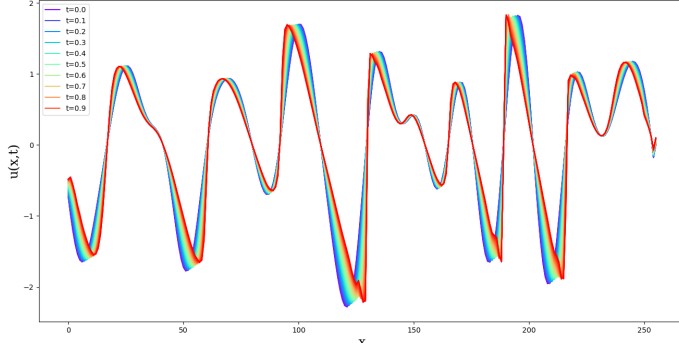

Figure 7: CoDA

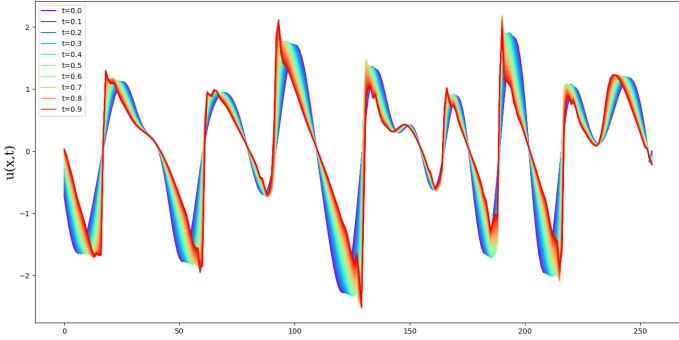

Figure 8: CAVIA

