# OpenReview forum: "LEARN TO ADAPT PARAMETRIC SOLVERS UNDER INCOMPLETE PHYSICS"
_ICLR.cc/2024/Workshop/AI4DiffEqtnsInSci — AI4DiffEqtnsInSci @ ICLR 2024 Poster_

### Official Review · Reviewer_RMcX · 2024-02-27
**Review of "Learn to adapt parametric solvers under incomplete physics"**

**Rating:** 8
**Confidence:** 4

**Review:**

The manuscript titled "Learn to adapt parametric solvers under incomplete physics" outlines an approach to predict the time dynamics of a dynamical system by incorporating known components of the physical characteristics and modeling only the unknown components. The paper is well written and clearly outlines the challenges, the approach used and the rationale for all decisions made towards the final proposed solution. The key idea is to decompose the RHS of a PDE into known and unknown components with an assumption on the functional form of this split and only train a model to learn the unknown dynamics. These unknown dynamics are further split into environment-agnostic and environment-dependent portions and a meta learning approach is described to be able to compute the environment-dependent portion of the model parameters using a linear model and learned low dimensional context vectors. This low dimensional context vector enables cost efficient adaptation to new environments. The experiments are conducted are described well and the results and ablations show the appeal of the proposed approach. With the exception of a couple of small details, all the conclusions drawn from the experiment results are well substantiated. Overall, I would recommend accepting this paper. Some specific comments:
- Section 1: "incorporating physical constraints within a loss function" Should cite Raissi et. al. 2019 here since they were the first to do this.
- Section 2.2: "Since the closure model operates on the low resolution dynamics model, a better decomposition is:" Why is composition a better approach for closure models? Typically closure terms are added as an additional source term, so wouldn't the residual approach be appropriate here as well?
- Eq 7: Is there any specific reasoning or intuition to choosing the sum of \theta^e and \theta^a vs. some other architectural mechanism to produce g_c(\theta^e, \theta^a)?
- Section 3: Citations for CoDA and CAVIA are needed here.
- Section 3: "We also extend CoDA into a hybrid version by adding to the original data-driven loss a physics-informed loss similar to the ones in PINNs" Should clarify if the physics-informed loss is applied on the whole model based on incomplete physics or a residual type approach with one part having the incomplete physics loss and the other one being purely data driven. The latter would be more comparable to the approach here and allow direct comparison of soft vs hard constraints, but the discussion makes it sound like the former was used.

---

### Official Review · Reviewer_WrDi · 2024-02-28
**A meta-learning approach for quick adaptation to new contexts in PDE learning**

**Rating:** 7
**Confidence:** 3

**Review:**

This paper proposes a meta-learning approach for learning parametrised PDEs, wherein the PDE is hardly encoded, as opposed to soft-constraint approaches where it is used in the loss. The proposed methodology allows for quicker adaptation to unseen contexts.

How do you compute $g_{p}$, i.e. the incomplete PDE part?

Also, references are missing for the CODA and CAVIA methods.

---

### Official Review · Reviewer_m3X6 · 2024-02-29
**Review of "LEARN TO ADAPT PARAMETRIC SOLVERS UNDER INCOMPLETE PHYSICS"**

**Rating:** 4
**Confidence:** 2

**Review:**

# Summary
The paper applies a meta-learning approach for efficiently adapting hybrid ODE and PDE models across multiple environments.

# Strengths
1. The problem studied by the paper, of generalizing hybrid models across environments, is interesting and impactful.
2. In general, the paper is easy to read and the figures, algorithms, and tables are well presented.
3. The range of experiments is good, and they appear to verify the effectiveness of the method.

# Weaknesses
1. As far as I can tell, the meta-learning strategy proposed here is identical to CoDA [1].
The primary contribution of this paper is therefore not the meta-learning strategy itself, but rather its application in a hybrid setting.
While applying CoDA to hybrid rather than purely data-driven models is a valid and interesting contribution, it nonetheless raises serious questions regarding the presentation of this paper.
In particular, **it would be easy to read the paper and think that the meta-learning strategy itself is a novel contribution**.
For example, the abstract states,
> ...we introduce a meta-learning strategy that captures context-specific variations inherent in each system, enhancing the model’s adaptability to generalize to new PDE parameters and initial conditions,

    while the introduction says,
    > To tackle this generalization problem, we propose a framework that can be viewed as a meta-learning approach for rapidly adapting the data-driven components of a hybrid mode [sic].

    In Section 2.3, "A Two Stage Framework", the paper's meta-learning algorithm is introduced *without any reference to CoDA*.
    In fact, CoDA is referenced only in the Appendix, where the authors state,
    > We propose a similar approach, but extends to more complex dynamical systems where partial knowledge is assumed.

    It's possible that I've missed some novel aspect of the proposed meta-learning algorithm other than its application to hybrid systems; if so, it still needs to be made very clear how the proposed algorithm relates to prior work, especially CoDA.
    As it stands, I unfortunately have serious doubts about the originality and the presentation of this work.
1. Both CoDA and CAVIA are used as baselines but without reference to the relevant papers, requiring some detective work by the reader.

# Other Comments
1. In many places, such as the abstract and introduction, the paper is written as if it were about PDEs only, but the experiments consider both ODEs and PDEs.
1. In the results in Table 1, it seems strange that "CoDA + Phys loss" performs strictly worse than CoDA, in many cases by multiple orders of magnitude.
If the physics loss were properly motivated, implemented, and trained, shouldn't it generalize at least as well as CoDA?

# Conclusion
Given my concerns about originality and presentation, outlined in the first weakness above, I cannot recommend this paper to be accepted in its current form.
In my opinion, the paper is presented as "a new meta-learning framework for hybrid models", while the reality is closer to "an existing meta-learning framework applied to hybrid models".
That being said, I still find the work relevant and interesting, and I look forward to seeing future versions.

# Citations

[1] Matthieu Kirchmeyer, Yuan Yin, Jeremie Dona, Nicolas Baskiotis, Alain Rakotomamonjy, and Patrick Gallinari. Generalizing to new physical systems via context-informed dynamics model. *International Conference on Machine Learning*, 2022.

---

### Meta-Review · Area_Chair_9MYg · 2024-03-01

**Recommendation:** Accept (Poster)

**Metareview:**

Authors propose a meta-learning approach to efficiently adapt to unseen context, applied to ODEs and PDEs. For the camera ready version, I recommend the following based on reviewer feedback: 1. citing similar prior work like CoDA 2. improving the novelty and clarity of the presentation and justify design choices. Authors should revise the framing to distinguish their contributions from prior meta-learning studies and provide more details on the model architecture and training methodology.

---

### Decision · Program_Chairs · 2024-03-02

Accept (Poster)